# MODEL-BASED ADVERSARIAL IMITATION LEARNING AS ONLINE FINE-TUNING

## ABSTRACT

In many real world applications of sequential decision-making problems, such as robotics or autonomous driving, expert-level data is available (or easily obtainable) with methods such as tele-operation. However, directly learning to copy these expert behaviours can result in poor performance due to distribution shift at deployment time. Adversarial imitation learning algorithms alleviate this issue by learning to match the expert state-action distribution through additional environment interactions. Such methods are built around standard reinforcement-learning algorithms with both model-based and model-free approaches. In this work we focus on the model-based approach and argue that algorithms developed for online RL are sub-optimal for the distribution matching problem. We theoretically justify utilizing conservative algorithms developed for the offline learning paradigm in online adversarial imitation learning and empirically demonstrate improved performance and safety on a complex long-range robot manipulation task, directly from images.

## 1 INTRODUCTION

Demonstrations are a natural way to teach to robots and intelligent agents, and can be obtained more easily than accurate than dense reward functions in the real world (i.e. via tele-operation) . Moreover, demonstrations alleviate many of the issues around exploration, which are prevalent in the online reinforcement learning setting. Behaviour Cloning (BC) (Pomerleau, 1988) is a classic algorithm which aims to copy expert behaviour from an available dataset of demonstrations. However, when deployed, it can suffer from instability (Ross et al., 2011) due to environment stochasticity, policy errors, or covariate shifts that distance the agent from the data support of the expert demonstrations. This issue can be alleviated through a wide enough expert coverage (Spencer et al., 2021), or the ability to query the expert (Ross et al., 2011). In contrast, inverse RL (IRL) (Finn et al., 2016b; Fu et al., 2018) and adversarial imitation learning (AIL) (Ho & Ermon, 2016; Finn et al., 2016a) aim to match the long-term state visitation distribution of the expert policy (Ghasemipour et al., 2019). Essentially, these methods use additional environment interactions to learn to self-correct when the agent deviates from the support of the expert distribution. AIL approaches formulate the imitation learning problem as a GAN (Goodfellow et al., 2014): a discriminator is trained to distinguish between the expert trajectories and those produced by the policy. The policy acts as a generator, producing rollouts from the environment, and is optimized with RL to fool the discriminator. Both model-free (Ho & Ermon, 2016; Kostrikov et al., 2019; Blondé & Kalousis, 2019) and model-based approaches (Baram et al., 2016; Rafailov et al., 2021) to the RL optimization problem have been developed. In general, these methods deploy a pre-existing policy optimization algorithm with the discriminator-based reward-learning framework described above.

We argue that algorithms designed for online RL are not well-suited to the imitation learning paradigm, as they carry out excessive exploration. In IRL/AIL settings, we already have samples from the expert distribution and the goal of the agent is to match that distribution, hence, we argue this setting is better suited as an offline-to-online fine-tuning problem. A recent imitation learning method, inverse Q-Learning (Garg et al., 2021), has drawn connections between imitation learning and conservative Q-learning (Kumar et al.) an offline RL method, and achieves good results in both the fully offline and the online learning settings. In this work we focus on model-based approaches. We theoretically justify the use of conservative model-based optimization used in offline RL for online imitation learning and propose a practical algorithm. We evaluate our method on the challenging

Franka Kitchen Environment with image observations only, and show a significant improvement in efficiency and stability over prior model-based algorithms. As far as we are aware, this is the most sample-efficient method to solve the environment, the first method to solve it directly from images, and the first method to solve it without access to a reward function.

## 2 Model-Based Adversarial Imitation Learning

Model-based algorithms for RL and IL involve learning an approximate dynamics model $\widehat{\mathcal{M}}$ using environment interactions. We then optimize a policy using large amount of data sampled from the learned model $\widehat{\mathcal{M}}$. We can bound the performance gap of the policy in the following theorem:

**Theorem 2.1.** *(Simultaneous policy and model deviation) Let $R_{\max} = \max_{(s,a)} r(s,a)$ be the maximum of the unknown reward in the MDP with unknown dynamics $\mathcal{M}$. For any policy $\pi$, we can bound the sub-optimality with respect to the expert policy $\pi^E$ as:*

$$\left| V_{\mathcal{M}}^{\pi^E} - V_{\mathcal{M}}^{\pi} \right| \leqslant \frac{R_{\max}}{1-\gamma} \underbrace{\mathbb{D}_{TV}(\rho_{\widehat{\mathcal{M}}}^{\pi}, \rho_{\mathcal{M}}^{E})}_{\text{distribution mathcing}} + \frac{\gamma \cdot R_{\max}}{(1-\gamma)^2} \underbrace{\mathbb{E}_{\rho_{\widehat{\mathcal{M}}}^{\pi}} \left[ \mathbb{D}_{TV}(\mathcal{M}(s'|s,a), \widehat{\mathcal{M}}(s'|s,a)) \right]}_{\text{model mismatch}} \quad (1)$$

*Proof.* We begin with the left-hand side

$$\left| V_{\mathcal{M}}^{\pi^E} - V_{\mathcal{M}}^{\pi} \right| \leqslant \underbrace{\left| V_{\mathcal{M}}^{\pi^E} - V_{\widehat{\mathcal{M}}}^{\pi} \right|}_{\text{Term I}} + \underbrace{\left| V_{\widehat{\mathcal{M}}}^{\pi} - V_{\mathcal{M}}^{\pi} \right|}_{\text{Term II}}$$

which is a simple application of the triangle inequality.

**Term I**: The first term yields the distribution matching component of Eq. 1 in the following way:

$$\left| V_{\mathcal{M}}^{\pi^E} - V_{\widehat{\mathcal{M}}}^{\pi} \right| = \frac{1}{1-\gamma} \left| \mathbb{E}_{s,a \sim \rho_{\mathcal{M}}^{\pi^E}}[r(s,a)] - \mathbb{E}_{s,a \sim \rho_{\widehat{\mathcal{M}}}^{\pi}}[r(s,a)] \right| \leqslant \frac{R_{max}}{1-\gamma} \mathbb{D}_{TV}(\rho_{\mathcal{M}}^{E}, \rho_{\widehat{\mathcal{M}}}^{\pi})$$

**Term II**: The second term yields the model mismatch part of the objective Eq. 1:

$$\left| V_{\widehat{\mathcal{M}}}^{\pi}(s_0) - V_{\mathcal{M}}^{\pi}(s_0) \right| \leqslant \gamma \left| \mathbb{E}_{\pi,\mathcal{M}}[V_{\mathcal{M}}^{\pi}(s_1)] - \mathbb{E}_{\pi,\widehat{\mathcal{M}}}[V_{\mathcal{M}}^{\pi}(s_1)] \right| + \gamma \left| \mathbb{E}_{\pi,\widehat{\mathcal{M}}} \left[ V_{\mathcal{M}}^{\pi}(s_1) - V_{\widehat{\mathcal{M}}}^{\pi}(s_1) \right] \right|$$

$$\leqslant \gamma \frac{R_{\max}}{1-\gamma} \mathbb{E}_{\pi} \left[ \mathbb{D}_{TV}(\mathcal{M}(s_1|s_0,a_0), \widehat{\mathcal{M}}(s_1|s_0,a_0)) \right] + \gamma \mathbb{E}_{\pi,\widehat{\mathcal{M}}} \left[ \left| V_{\mathcal{M}}^{\pi}(s_1) - V_{\widehat{\mathcal{M}}}^{\pi}(s_1) \right| \right]$$

The first inequality is a direct application of the triangle inequality. The second line follows from the same considerations we used to bound **Term I** above. We can then recursively apply the same reasoning to the final term to obtain the model mismatch component of the bound. $\square$

In a prior work (Rafailov et al., 2021) the authors consider a similar objective to Eq. 1, however they assume a uniform model discrepancy bound and only optimize the first term of the inequality. Several prior works (Yu et al., 2020; Kidambi et al., 2020; Rafailov et al., 2020) have considered the model mismatch objective of Eq. 1 in the context of offline model-based RL, while we consider the online adversarial imitation learning problem.

## 3 Our Method

Our full algorithm has several components: 1) variational dynamics model, 2) state-action discriminator, 3) critic, and 4) actor policy. We will discuss all of these in more detail. During the online training phase, we iterate between training the model, discriminator, actor and critic.

**Variational Dynamics Models** We use use a recurrent state-space model (RSSM) (Hafner et al., 2019; 2020), but without reward prediction and optimize the standard ELBO bound:

$$\mathbb{E}_{q_\theta} \left[ \sum_{t=1}^{\tau} \underbrace{\log p_\theta(x_t|s_t)}_{\text{reconstruction}} - \underbrace{\mathbb{D}_{KL}(q_\theta(s_t|x_t, s_{t-1}, a_{t-1}) || \mathcal{M}_\theta(s_t|s_{t-1}, a_{t-1}))}_{\text{latent forward model}} \right] \quad (2)$$

Here both the inference distribution $q_\theta$ and the latent dynamics model $\mathcal{M}_\theta$ are Gaussian distributions parameterized by neural networks. Following (Rafailov et al., 2020) we train a latent ensemble of dynamics models $\{\mathcal{M}_{\theta^i}\}_{i=1}^K$ by selecting a different member of the ensemble to evaluate the above loss at every time step in the trajectory.

**Reward Formulation** Directly applying Theorem 1 in (Rafailov et al., 2021), we can bound the objective of Eq. 1 by optimizing the same bound in the learned models' latent belief space. In more detail, we consider sequences of data of the form $\tau = (\boldsymbol{x}_{1:T}, \boldsymbol{a}_{1:T})$. At each agent training step, we infer latent states $\boldsymbol{s}_{1:T}^0 \sim q_\theta(\boldsymbol{s}_{1:T}|\boldsymbol{x}_{1:T}, \boldsymbol{a}_{1:T})$. We also denote $\boldsymbol{a}_{1:T}$ as $\boldsymbol{a}_{1:T}^0$. Using these states as starting points, we use the policy $\pi_\psi$ to generate $H$-step rollouts steps with the following notation: $\hat{\boldsymbol{a}}_j^t \sim \pi_\psi(\boldsymbol{a}|\hat{\boldsymbol{s}}_j^t)$ and $\hat{\boldsymbol{s}}_j^{t+1} \sim p_\theta(\boldsymbol{s}|\hat{\boldsymbol{a}}_j^t, \hat{\boldsymbol{s}}_j^t)$. Following standard off-policy learning algorithms, we use critics $\{Q_{\psi^i}\}_{i=1}^m$ and and target networks $\{\bar{Q}_{\psi^i}\}_{i=1}^m$.

We can bound the distribution matching component of Eq. 1 through Pinsker's inequality and follow a standard adversarial approach by training a discriminator (Ghasemipour et al., 2019):

$$\min_{D_\psi} -\frac{1}{T}\sum_{i=1}^{T}\log D_\psi(\boldsymbol{s}_i^E, \boldsymbol{a}_i^E) - \frac{1}{HT}\sum_{i=1,t=0}^{T,H-1}\log(1 - D_\psi(\boldsymbol{s}_i^t, \boldsymbol{a}_i^t)) \tag{3}$$

where $\boldsymbol{s}_{1:T}^E \sim q_\theta(\boldsymbol{s}_{1:T}|\boldsymbol{x}_{1:T}^T, \boldsymbol{a}_{1:T}^E)$ are the expert's inferred latent states and actions. On-policy samples from the model also give theoretical justification for this discriminator learning objective as the expectation is taken under the current policy. We cannot directly optimize the model mismatch component of Eq. 1, as we cannot directly estimate divergence factor. Instead, following prior work (Yu et al., 2020; Kidambi et al., 2020; Rafailov et al., 2020), we optimize a surrogate objective based on ensemble model disagreement. In particular, we use

$$\mathbb{E}_{\rho_{\widehat{\mathcal{M}}}^\pi}\left[\mathbb{D}_{TV}(\mathcal{M}(s'|s,a), \widehat{\mathcal{M}}(s'|s,a)\right] \approx \frac{1}{HT}\sum_{i=1,t=0}^{T,H-1}\left[\text{std}(\{\mu_{\theta^i}(\hat{\boldsymbol{s}}_i^t, \hat{\boldsymbol{a}}_i^t)\}_{i=1}^K)\right]$$

where $\mu_{\theta^i}(\boldsymbol{s}, \boldsymbol{a})$ is the parameterization of the mean of $i-$th Gaussian latent model $\mathcal{M}_{\theta^i}(\cdot|\boldsymbol{s}, \boldsymbol{a})$. The combined final reward is then:

$$\boldsymbol{r}(\boldsymbol{s}, \boldsymbol{a}) = \log D_\psi(\boldsymbol{s}, \boldsymbol{a}) - \log(1 - D_\psi(\boldsymbol{s}, \boldsymbol{a})) - \alpha\text{std}(\{\mu_{\theta^i}(\boldsymbol{s}, \boldsymbol{a})\}_{i=1}^K) \tag{4}$$

where $\alpha$ is a tunable hyper-parameter. We note that this still a fully differentiable function of the state. We can then label all data and model-sampled latent transitions $\boldsymbol{r}_i^t = \boldsymbol{r}(\boldsymbol{s}_i^t, \boldsymbol{a}_i^t)$.

**Actor Optimization** Once we have rewards, we can calculate Monte-Carlo based policy returns:

$$V_0^{\pi_\psi}(\hat{\boldsymbol{s}}_j^t) = \min_{i=1:m}\{Q_{\psi^i}(\hat{\boldsymbol{s}}_j^t, \hat{\boldsymbol{a}}_j^t)\}$$

$$V_K^{\pi_\psi}(\hat{\boldsymbol{s}}_j^t) = \sum_{k=1}^{K}\gamma^{k-1}\hat{\boldsymbol{r}}_j^{k+t} + \gamma^K V_0^{\pi_\psi}(\hat{\boldsymbol{s}}_j^{t+K})$$

And compute the standard TD($\lambda$) estimate:

$$V^{\pi_\psi}(\hat{\boldsymbol{s}}_j^t) = (1-\lambda)\sum_{k=1}^{H-t-1}\lambda^{k-1}V_k^{\pi_\psi}(\hat{\boldsymbol{s}}_j^t) + \lambda^{H-t-1}V_{H-t}^{\pi_\psi}(\hat{\boldsymbol{s}}_j^t) \tag{5}$$

And the final actor objective is:

$$\mathcal{L}_{\pi_\psi}^{\text{model}} = -\frac{1}{HT}\left[\sum_{t=0,j=1}^{H-1,T}\lambda V^{\pi_\psi}(\hat{\boldsymbol{s}}_j^t) + (1-\lambda)V_0^{\pi_\psi}(\hat{\boldsymbol{s}}_j^t)\right] \tag{6}$$

which maximizes the expected MC return at the dataset and rollout states together. Notice that this fully differentiable function of the policy parameters, by differentiating through the model and reward function.

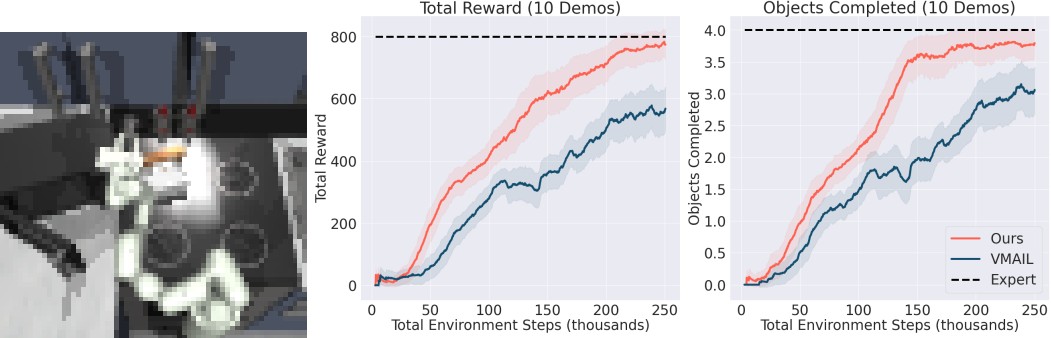

Figure 1: Left: Observation from the Franka Kitchen Environment. Right: Performance of our algorithm compared to a straightforward model-based imitation approach. Our algorithm learns faster and in a more stable manner than VMAIL.

**Critic Optimization** We can use MC return estimates to train the critics as well. We recompute the critic target values $\bar{V}^k(\hat{s}_j^t)$ for all states similarly to Eq. 5 using the target networks $\{\bar{Q}_{\psi i}\}_{i=1}^m$. Critics are trained on both the model-generated and real data with two losses:

$$\mathcal{L}_{Q_{\psi i}}^{\text{model}} = \frac{1}{HT}\left[\sum_{t=0,j=1}^{H-1,T}(Q_{\psi i}(\hat{s}_j^t, \hat{a}_j^t) - \bar{V}^{\pi_\psi}(\hat{s}_j^t))^2\right] \qquad (7)$$

$$\mathcal{L}_{Q_{\psi i}}^{\text{data}} = \frac{1}{T-1}\left[\sum_{j=1}^{T-1}\left(Q_{\psi i}(s_j^0, a_j^0) - (\hat{r}_{j+1}^0 + \gamma((1-\lambda)\bar{V}_0^{\pi_\psi}(s_{j+1}^0) + \lambda\bar{V}^{\pi_\psi}(s_{j+1}^0)))\right)^2\right] \qquad (8)$$

The final critic loss is a combination of the two losses:

$$\mathcal{L}_{Q_{\psi i}} = \mathcal{L}_{Q_{\psi i}}^{\text{model}} + \mathcal{L}_{Q_{\psi i}}^{\text{data}} \qquad (9)$$

The loss $\mathcal{L}_{Q_{\psi i}}^{\text{data}}$ is computed on transitions sampled form the dataset trajectories through the inference model $q_\theta$. Training the critics on the available expert demonstrations serves as a strong supervision.

## 4 EXPERIMENTS

We evaluate our method on the Franka Kitchen environment (Gupta et al., 2019; Fu et al., 2020), shown in Fig, 1. The environment consist of a Franka robot with a joint-space control in a kitchen setting. The observation consists of a single RGB image and we do not assume access to robot or object states. The agent is provided with 10 demonstrations of the microwave, kettle, light switch, slide cabinet task configuration. We compare our method to VMAIL (Rafailov et al., 2021) an online model-based imitation learning approach that does not employ conservatism. Training results are shown in Fig. 1 we see that our method learns faster and in a more stable way as compared to VMAIL. As far as we are aware, this is the most sample-efficient method to solve the environment, the first method to solve it directly from images, and the first method to solve it without access to a reward function.

## 5 CONCLUSION

In this work we argue that policy optimization algorithms designed for online RL are not well suited to the IRL/AIL setting as they carry out excessive exploration. We focus on the model-based case and argue that conservative models used for offline RL are better suited for policy optimization. We provide a theoretical justification for our design choices, as well as a practical algorithm and evaluate it on a challenging robot manipulation task. The proposed algorithm achieves faster and more stable performance as compared to previous model-based imitation learning approaches. In future work we plan to evaluate our method on further domains, as well as the fully offline IL setting.

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

## A    APPENDIX

You may include other additional sections here.

