# OpenReview forum: "Model-Based Adversarial Imitation Learning As Online Fine-Tuning"
_ICLR.cc/2023/Workshop/RRL — RRL 2023 Poster_

### Official Review · Reviewer_smkN · 2023-02-28
**Decent contribution, but could use some polishing**

**Rating:** 2
**Confidence:** 1

**Review:**

Unfortunately, I only had time for a superficial review. My apologies.

**Summary**: In adversarial imitation learning, a policy is trained to fool a discriminator, using any online RL algorithm for training. The authors argue that this is not a good practice, because online RL algorithms are designed to explore a lot. Instead, they take inspiration from conservative Q-Learning (CQL) and propose a conservative model-based RL algorithm for online adversarial imitation learning. They demonstrate it on the Franka Kitchen Environment.

**Quality**: I did not work through all the details, but my impression is that the authors make a valid point and their algorithm makes sense. The demonstration also appears reasonable.

**Clarity**: What could be improved in the paper, even for a short workshop paper, is the presentation. The introduction is well-written, but the theory part is hard to read: symbols are not defined, assumptions are not stated, and the consequence of statements is not always clear.

**Significance**: The paper topic is a clear match for the workshop. While the paper is rather iterative and follows in the footsteps of several other works, for a workshop I consider its contribution to be significant enough.

**Bottom line**: A fine contribution that could use a bit more polishing for the camera-ready version.

**Details**:
- Page 1, first paragraph of main paper, beginning: typo "more easily than accurate than dense"
- Theorem 2.1: What are the assumptions here? For instance, are rewards assumed to be non-negative? Also, please define all symbols.
- Page 2, just above Eq. (2): Typo "use use"
- Appendix: The authors may want to remove the dummy appendix. Or, you know, replace it with actual details about their theory and experiments :)

---

### Official Review · Reviewer_iMFC · 2023-03-02
**Well-illustrated work on offline model-based algorithm for adversarial imitation learning**

**Rating:** 3
**Confidence:** 3

**Review:**

The paper is focused on how model-based RL algorithms benefit adversarial imitation learning. Experiments show that the result is promising compared to the baseline. In this review, we list the strengths and weaknesses of the work.

The topic fits the guideline of Reincarcinating RL. The background and the related work parts are detailed and help illustrate the innovative insight of the work. The method part is well illustrated with detailed proof and formula. The experiment comparing the method with the baseline VMAIL looks promising, but there can be more baseline experiments and maybe extend to other environments.

In summary, the paper works well in the Franka kitchen domain compared with one baseline method and illustrates the efficiency of offline learning algorithms on adversarial imitation learning.